# Proteomic Characterization of PAMs with PRRSV-ADE Infection

**DOI:** 10.3390/v15010036

**Published:** 2022-12-22

**Authors:** Pengli Xu, Wen Li, Shijie Zhao, Zhiying Cui, Yu Chen, Yi-na Zhang, Jing Chen, Pingan Xia

**Affiliations:** 1College of Veterinary Medicine, Henan Agricultural University, Longzi Lake 15#, Zhengzhou 450046, China; 2College of Life Science, Henan Agricultural University, Longzi Lake 15#, Zhengzhou 450046, China

**Keywords:** PRRSV, PRRSV-ADE infection, PAMs, proteomics, virus–antibody–host interactions

## Abstract

The antibody-dependent enhancement (ADE) effect of a PRRSV infection is that the preexisting sub- or non-neutralizing antibodies specific against PRRSV can facilitate the virus entry and replication, and it is likely to be a great obstacle for the selection of immune strategies and the development of high-efficiency PRRSV vaccines. However, the proteomic characterization of primary alveolar macrophages (PAMs) with a PRRSV-ADE infection has not yet been investigated so far. Therefore, we performed a tandem mass tag (TMT)-based quantitative proteomic analysis of PAMs with a PRRSV-ADE infection in this study. The results showed that a total of 3935 differentially expressed proteins (DEPs) were identified in the PAMs infected with PRRSV-ADE, including 2004 up-regulated proteins and 1931 down-regulated proteins. Further, the bioinformatics analysis for these DEPs revealed that a PRRSV-ADE infection might disturb the functions of ribosome, proteasome and mitochondria. Interestingly, we also found that the expression of the key molecules in the innate immune pathways and antiviral proteins were significantly down-regulated during a PRRSV-ADE infection. This study was the first attempt to analyze the proteomic characterization of PAMs with a PRRSV-ADE infection in vitro. Additionally, the findings will provide valuable information for a better understanding of the mechanism of virus–antibody–host interactions during a PRRSV-ADE infection.

## 1. Introduction

Porcine reproductive and respiratory syndrome virus (PRRSV) can cause severe reproductive failure in pregnant sows and respiratory disease in pigs of all ages and causes incalculable economic losses to the global pig economy every year. Although vaccination is the most effective method for preventing a PRRSV infection, the effects of the commercially available PRRSV vaccines are unsatisfactory. Except for the genetic diversity of PRRSV itself, the antibody-dependent enhancement (ADE) effect may be another major factor reducing the clinical immune effect of commercial PRRSV vaccines, which was that the preexisting non-neutralizing antibody or sub-neutralizing antibody could enhance the infection and replication of the virus [1,2,3]. 

In 1992, Christianson et al. firstly found the ADE effect of a PRRSV infection in primary porcine alveolar macrophages (PAMs), and then this phenomenon was also confirmed in vivo and in vitro by Yoon et al. [2,4,5]. Moreover, Yoon et al. found that a PRRSV-ADE infection caused more severe clinical symptoms in North American swine herds challenged with PRRSV vaccines [5]. Previous studies have shown that the PRRSV-ADE infection was principally mediated by Fc receptors (Fc Rs) including Fcγ RI, Fcγ RIIb, Fcγ RIII and Fcε RI [6,7,8]. Furthermore, the activation of these Fc Rs could differently regulate the innate immune response to promote virus replication during a PRRSV-ADE infection. However, the molecular mechanisms of other Fc Rs-mediated PRRSV-ADE infection are yet to be precisely elucidated, except for FcγR IIb. Moreover, Chan et al. reported that a dengue virus (DENV)-ADE infection could enhance the infection and replication of a virus through modulating the expression of host dependency factors, mitochondrial respiratory chain complexes and ribosomal genes [9]. Therefore, a deeper investigation into the molecular mechanism of a PRRSV-ADE infection should be implemented.

At present, different proteomic approaches have also been applied to explore the molecular mechanisms of a PRRSV infection in vivo or in vitro [10,11,12]. These researches could contribute to better understanding the pathogenic mechanisms of a PRRSV infection, but there are limited studies about the connection between a PRRSV-ADE infection and the host cells. Therefore, we first profiled the tandem mass tag (TMT)-based quantitative proteomic analysis of PAMs in the early period of a PRRSV-ADE infection in this study. It will be helpful for elucidating the molecular mechanism of virus–antibody–host interactions during the PRRSV-ADE infection.

## 2. Materials and Methods

### 2.1. Cells, Virus and Antibodies

The primary PAMs obtained from PRRSV-negative pigs of four to eight weeks old were maintained in the Roswell Park Memorial Institute (RPMI)-1640 medium (Solarbio, Beijing, China) supplemented with 10% (v/v) fetal bovine serum (FBS) (Excell, Shanghai, China). The North American-type PRRSV strain HeN-3 (GenBank ID: ON645930) was isolated and stored in our laboratory, and its 50% tissue culture infectious dose (TCID_50_) was determined in MARC-145 cells by the Reed–Muench method. The pig anti-PRRSV IgG (Enzymelinked immunosorbent assay (ELISA) antibody titer: 6400) and pig-negative IgG were purified by Diethylaminoethyl (DEAE) cellulose chromatography and stored in our laboratory.

### 2.2. PRRSV-ADE Infection Assay in PAMs

The PAMs were prepared in a 24-well plate at a ratio of 5 × 10^5^ cells/well. After attachment for 12 h, PRRSV (2000 TCID_50_/mL)/anti-PRRSV IgG (400 μg/mL) immune complexes (PRRSV-ICs) and PRRSV (2000 TCID_50_/mL)/porcine negative IgG (400 μg/mL) admixtures (PRRSV-NI) were prepared as previously described [6]. Then, the adherent PAMs in the plate were, respectively, inoculated with PRRSV-ICs and PRRSV-NI, and normal cells were used as a mock trial. After 2 h, the medium was changed to the fresh RPMI-1640 medium containing 10% FBS. The cells and culture supernatants of all groups were, respectively, collected to detect the proliferation of PRRSV at 24 h post-infection (hpi) by an immunofluorescence assay (IFA) or quantitative reverse transcription PCR (RT-qPCR). 

### 2.3. Sample Processing for TMT-Based Proteomics

To explore the proteomic characterization of PAMs in the early period of a PRRSV-ADE infection, the cells of the PRRSV-ICs group and the PRRSV-NI group at 6 hpi were dissolved in lysis solution with SDT buffer (4% (*w*/*v*) SDS, 100 mM Tris-HCl pH 7.6, 0.1 M DTT) to extract the protein. Then, all the protein samples were processed as follows: filter-aided sample preparation, TMT labeling and reversed-phase peptide fractionating.

### 2.4. LC-MS/MS Analysis

The fractions were separated using the Easy nLC HPLC system (Thermo Fisher Scientific, Waltham, MA, USA). The samples were loaded onto a reverse phase trap column (Acclaim PepMap100, 100 μm × 2 cm, nanoViper C18) (Thermo Fisher Scientific, Waltham, MA, USA), and separated by the analytical column (EASY column, 75 μm × 10 cm, 3 μm, C18-A2) (Thermo Fisher Scientific, Waltham, MA, USA). Chromatographic separation was performed with a linear gradient of buffer A (0.1% formic acid) and buffer B (84% acetonitrile with 0.1% formic acid) for 60 min at the flow rate of 300 nL/min controlled by IntelliFlow technology (0–55% buffer B for 80 min, 55–100% buffer B for 5 min, and holding in 100% buffer B for 5 min). Finally, the samples were subjected to the mass spectrometer Q-Exactive (Thermo Fisher Scientific, Waltham, MA, USA) in a positive ion mode. The precursor ions of MS1 were scanned in the range of 300–1800 *m*/*z* and the resolution was 70,000 at *m*/*z* 200. The precursor ions were selected to acquire the product ions in higher energy collision dissociation (HCD) mode, with the normalized collision energy of 30 eV. The resolution of MS2 was 17,500 at *m*/*z* 200 and the underfill was defined as 0.1%.

The MS raw data were processed by the MASCOT engine (Matrix Science, London, UK, version 2.2) and the Proteome Discoverer (version 1.4) for the identification and quantitation of the proteins. The retrieval parameters were set as follows: the cleavage enzyme was Trypsin, the max missed cleavage was set as 2, the peptide mass tolerance was 20 ppm, the fragment mass tolerance was set as 0.1 Da, the fixed modification was carbamidomethyl and the variable modification was oxidation. The Sus_scrofa database was downloaded from UniProtKB (www.uniprot.org, accessed on 25 January 2021) and a false discovery rate (FDR) < 1% was used for the identification. The quantification of a protein was calculated by the median of only the unique peptides of the protein.

### 2.5. Bioinformatics Analysis

In this study, the original data were searched through the Mascot (version 2.2), and then the retrieved results were imported into the Proteome Discoverer (version 1.4) for the identification and quantitation analysis. CELLO (http://cello.life.nctu.edu.tw/, accessed on 25 January 2021) was used for the subcellular localization prediction of the differentially expressed proteins (DEPs). Gene ontology (GO) analysis and the Kyoto Encyclopedia of Genes and Genomes (KEGG) pathway annotation of all the DEPs were performed using Blast2GO (https://www.blast2go.com/, 25 January 2021) and using the KAAS (KEGG Automatic Annotation Server) software (http://www.genome.jp/kegg/, accessed on 25 January 2021) [13]. Moreover, the enrichment analysis of the DEPs for the GO annotation and KEGG pathway annotation were performed based on Fisher’s exact test (*p* < 0.05). The interaction relationships between the target proteins was based on the STRING database (http://string-db.org/, accessed on 25 January 2021) and the result was visualized by the Cytoscape (version 3.9.1).

### 2.6. Immunofluorescence Assay (IFA)

The cells of all groups were, respectively, fixed with 4% paraformaldehyde and this was followed by a treatment with 0.1% Triton-X100 for 10 min. The cells were closed with 5% BSA at room temperature for 2 h and were then washed three times with PBST. The cells were incubated with an anti-PRRSV N mouse monoclonal antibody for 1 h. The cells were washed three times with PBST and incubated with Alexa FluorTM 546 goat anti-mouse IgG (H + L) (Invitrogen, NY, USA) for 1 h in the dark, and then dyed for 10 min with DAPI. Finally, these images were observed and taken by the fluorescence microscopy.

### 2.7. RT-qPCR Assay

The total RNAs were extracted from the cells of the PRRSV-NI group and the PRRSV-ICs group at 6 hpi, and reverse-transcribed to cDNA by using a HiScript II Q RT SuperMix (Vazyme, Nanjing, China). Then, all the cDNA samples were detected by RT-qPCR, and each cDNA sample was performed in triplicate. The expression levels of the target proteins for two groups were estimated using the 2^−ΔΔCt^ method. All RT-qPCR primers used in this study are listed in Appendix A.

### 2.8. Western Blot Analysis

The cells of the PRRSV-NI group and the PRRSV-ICs group at 6 hpi were lysed by the cell lysis buffer containing 1 mM of PMSF (Beyotime, Shanghai, China), and, respectively, mixed with 5 × SDS-PAGE loading buffer. The samples were separated by 8% SDS-PAGE and were then transferred to the Nitrocellulose Blotting Membrane (GE Healthcare, Boston, MA, USA). The membranes were blocked with 5% BSA at room temperature for 2 h and were then incubated overnight at 4 °C with the primary antibodies, including the TANK-binding kinase 1 (TBK-1) (Cell Signaling Technology, Boston, MA, USA), signal transducer and the activator of transcription 1 (STAT-1) (Cell Signaling Technology, Boston, MA, USA), signal transducer and activator of transcription 2 (STAT-2) (Cell Signaling Technology, Boston, MA, USA) and Myxovirus resistance protein 1 (Mx1) (Santa Cruz, Dallas, TX, USA). After being washed three times with PBST, the membranes were incubated with horseradish peroxidase (HRP)-conjugated goat anti-rabbit IgG (Cell Signaling Technology, Boston, MA, USA) or HRP-conjugated goat anti-mouse IgG (Cell Signaling Technology, Boston, MA, USA) at room temperature for 1 h. Finally, the signals were detected using the GE AI680 imaging system (GE Healthcare, Boston, MA, USA).

### 2.9. Statistical Analysis

The data were analyzed by the GraphPad Prism software (version 6.0) using Student’s *t*-test. A *p* value < 0.05 was considered significant. The degree of statistical significance is indicated by asterisks (*** *p* < 0.001, ** *p* < 0.01, * *p* < 0.05).

## 3. Results

### 3.1. Establishment and Verification of a Cell Model with PRRSV-ADE Infection 

In order to perform the proteomic analysis of the PAMs with a PRRSV-ADE infection, we initially established a PRRSV-ADE infected cell model and identified the virus propagation in the PAMs with a PRRSV-ADE infection by RT-qPCR and IFA. The results of RT-qPCR showed that the viral load of the cell culture supernatant in the PRRSV-ICs group was significantly higher at 24 hpi, compared with the PRRSV-NI group. Additionally, the PRRSV replication in the PAMs of the mock group was not detected (Figure 1A). Furthermore, the results of the IFA showed that the specific fluorescent signals of the PRRSV N protein in the PAMs of the PRRSV-ICs group were significantly more than the PRRSV-NI group at 24 hpi, and it was not found in the mock group (Figure 1B). Therefore, these data indicated that the cell model could be used for the further proteomic analysis of the PAMs with a PRRSV-ADE infection.

### 3.2. Analysis of the Differentially Expressed Proteins in the PAMs with PRRSV-ADE Infection

In this study, 6858 proteins were identified and 6813 of which were successfully quantified in the PAMs of the PRRSV-ICs group and the PRRSV-NI group. Based on the *t*-test (the cut-offs as a *p*-value < 0.05 and a fold change >1.2 or <0.83), 3935 DEPs were screened in the PAMs infected with PRRSV-ADE, of which 2004 were significantly up-regulated and 1931 were significantly down-regulated (Figure 2). 

### 3.3. Bioinformatics Analysis of the DEPs

First, the subcellular localization analysis showed that most of these DEPs were localized in the nuclear, cytoplasmic, mitochondrial, plasma-membrane and extracellular. Significantly, the up-regulated proteins were mainly localized in the nuclear, mitochondrial, plasma-membrane cytoplasmic and extracellular. While most of the down-regulated proteins were localized in the nuclear and cytoplasmic in PAMs with a PRRSV-ADE infection (Figure 3).

Second, the results showed that these DEPs were involved in three distinctive functional sets: the biological process (BP), molecular function (MF) and cellular component (CC). As shown in Figure 4A, a PRRSV-ADE infection was mainly involved in the organonitrogen compound biosynthetic process, cellular amide metabolic process, small molecule metabolic process, organonitrogen compound metabolic process, peptide metabolic process and amide biosynthetic process in BP. Moreover, a PRRSV-ADE infection mainly effected the three molecular functions: the catalytic activity, oxidoreductase activity and structural constituent of ribosome (Figure 4B). In terms of the CC annotation, most of these DEPs were primarily localized in the cytoplasm, mitochondrion and ribosome in PAMs infected with PRRSV-ADE (Figure 4C).

In order to more systematically explore the mechanism of a PRRSV-ADE infection, we have investigated the changes in the KEGG pathways. The 3935 DEPs were involved in the 222 KEGG pathways, and the 26 KEGG pathways were significantly enriched by Fisher’s exact test (*p* < 0.05). Interestingly, the top 26 KEGG pathways were divided into six clusters: the organismal systems, metabolism, human diseases, genetic information processing, environmental information processing and cellular processes. Moreover, the enrichment analysis of the top 26 KEGG pathways revealed that there was a higher correlation between a PRRSV-ADE infection and the top 6 KEGG pathways including ribosome, oxidative phosphorylation, diabetic cardiomyopathy, thermogenesis, proteasome and glycolysis/gluconeogenesis, than the other 20 KEGG pathways (Figure 5). 

### 3.4. Analysis of the Protein–Protein Interactions (PPI) Network

To investigate the PPI network of these DEPs enriched in the top 6 KEGG pathways, 143 DEPs were picked and the PPI networks between them were analyzed by the STRING database. As shown in Figure 6, 1130 PPIs were detected from these DEPs using the high confidence mode (minimum required interaction score ≥0.9. PPI enrichment *p* < 1.0 × 10^−16^). Significantly, the DEPs centrally distributed in three PPI network clusters, and they were mainly involved in ribosome, proteasome and mitochondria. In conclusion, the results indicated that an PRRSV-ADE infection might differentially regulate the functions of ribosome, proteasome and mitochondria. 

### 3.5. Analysis of the DEPs Associated with Antiviral Innate Immunity

Previous studies have shown that a PRRSV-ADE infection could markedly suppress the antiviral innate immunity, but the mechanisms were rarely researched. Intriguingly, the KEGG pathways analysis of all the DEPs also revealed that four signaling pathways have changed in PAMs with a PRRSV-ADE infection, including the toll-like receptor signaling pathway, the NOD-like receptor signaling pathway, the RIG-I-like receptor signaling pathway and the JAK-STAT signaling pathway. Therefore, to fully understand the changes in the antiviral innate immune response in PAMs with a PRRSV-ADE infection, we analyzed all the DEPs and found that 40 DEPs were involved in the antiviral innate immunity in this study. As shown in Table 1, the expression of pattern recognition receptors (PRRs) was regulated differently in PAMs with a PRRSV-ADE infection. Among these PRRs, the expression of TLR2, TLR3, TLR4, TLR8, DHX9 and DHX15 were significantly increased, and the expression of others (DHX29, DHX36, DHX58, DDX3X, DDX6 and ZNFX-1) were significantly decreased. Significantly, the expression levels of the key upstream molecules of the IFN signaling (MyD88, TBK-1, TRAF3 and IRF7), the key molecules of the JAK-STAT signaling pathway (STAT-1, STAT-2, STAT-3, STAT-6 and IRF9) and antiviral proteins (IFIT1, IFIT2, IFIT3, IFIT5, Mx1, TRIM21, TRIM25, TRIM26, TRIM34, RSAD2 and OAS1) were significantly inhibited in the PAMs of the PRRSV-ICs group, compared with the PRRSV-NI group. Furthermore, the expression of several innate immune response regulators (COX-5B, MMP-9, TRIM21, TRIM25 and TRIM26) in the PAMs were also regulated differently during the PRRSV-ADE infection.

### 3.6. Validation of Results from the DEPs by RT-qPCR or Western Blot

In order to verify the reliability of the MS results, the analysis of mRNA levels and/or expression levels for 9 DEPs randomly chosen from the proteomic data were performed by RT-qPCR and Western blot. The results showed that compared with the PRRSV-NI group, the mRNA levels of four proteins including *OAS1*, *ISG56*, *Mx1* and *RSAD2* were dramatically down-regulated in the PAMs of the PRRSV-ICs group, while that of the *COX-5B* and *MMP-9* were dramatically up-regulated (Figure 7A). Meanwhile, the expression levels of four proteins, including TBK-1, STAT-1, STAT-2 and Mx1, were significantly down-regulated by Western blotting in the PAMs of the PRRSV-ICs group (Figure 7B). These data were consistent with the changes in the expression observed in the TMT-based quantitative proteomic analysis.

## 4. Discussion

In this study, we initially have established a PRRSV-ADE infected cell model, and then used the TMT-labeled LC-MS/MS to analyze the proteome of PAMs in the early stage of a PRRSV-ADE infection. The results showed that a total of 6813 proteins were obtained, and 3935 DEPs (2004 up-regulated proteins and 1931 down-regulated proteins) were identified based on the *t*-test (the basis of a fold change > 1.2 or < 0.83 and *p* < 0.05). The RT-qPCR or/and immunoblotting results of those randomly selected DEPs were also consistent with the proteomic analysis in this study. The analysis of the GO annotation, KEGG pathways annotation and PPI network for all the DEPs revealed that a PRRSV-ADE infection mainly affected the functions of the cellular components, such as the ribosome, proteasome and mitochondria. Moreover, we also found that the innate immune signals and the expression of multiple antiviral proteins were observably inhibited in the PAMs during the PRRSV-ADE infection. The findings in this study are discussed below.

### 4.1. PRRSV-ADE Infection Inhibits the Innate Immune Signals

As shown in Table 1, our results showed that the expression of TLR2, TLR3, TLR4 and TLR8 were markedly up-regulated and the expression of other RNA sensors (DHX9, DHX15, DHX29, DHX36, DHX58, ZNFX-1, DDX1, DDX3X and DDX6) were significantly down-regulated in the PAMs with a PRRSV-ADE infection. Multiple studies reported that these viral RNA sensors could also specifically sense pathogenic RNA to stimulate the TIR-domain-containing adapter-inducing interferon-β (TRIF)- or the mitochondrial antiviral signaling protein (MAVS)-mediated type I IFN signaling, the type III IFN signaling and the pro-inflammatory cytokines signaling, except for the TLRs [14,15,16,17,18]. Furthermore, we also found that COX-5B was significantly up-regulated, while TRIM25, TRIM21, TRIM27 and TRIM26 were significantly down-regulated in the PAMs with a PRRSV-ADE infection in this study. Among them, COX-5B represses the MAVS aggregation to hinder the activation of the innate immune signaling, but TRIM25, TRIM21 and TRIM26 participate in positively regulating the type I IFN pathway by interacting with some key molecules including RIG-I, MAVS, TBK-1 or IRF3 [19,20]. In addition, Wan et al. also reported that Fcγ RIIb reduced the phosphorylation of TBK-1 and inhibited the expression of IFN-β by recruiting Src homology 2-containing inositol phosphatase-1 (SHIP-1) during the PRRSV-ADE infection [7]. Significantly, our results showed that the expression of six major adaptor molecules (Myd88, IRAK2, IRAK4, TRAF3, TBK-1 and IRF7) downstream of RNA sensors were dramatically decreased in the PAMs with a PRRSV-ADE infection. Meanwhile, the studies of Zhang et al. and Bao et al. showed that a PRRSV-ADE infection could inhibit the production of IFN-α, IFN-β, IFN-λ and TNF-α [6,7,21]. Thus, these results demonstrated that a PRRSV-ADE infection might roundly inhibit the pattern recognition receptor (PRR)-mediated innate immune signals to facilitate a viral proliferation through multiple strategies. 

### 4.2. PRRSV-ADE Infection Down-Regulates the Expression of Antiviral Proteins

We found that the expression of several key adapter proteins (IRF9, STAT-1, STAT-2, STAT-3 and STAT-6) of the JAK-STAT signaling pathway were significantly down-regulated, and MMP-9 were also significantly up-regulated in the PAMs with a PRRSV-ADE infection (Table 1). MMP-9 can facilitate virus replication through intercept IFN/JAK-STAT signals [22]. Meanwhile, multiple studies reported that the IFN signals were dramatically down-regulated during the PRRSV-ADE infection [6,7,21]. It indicates that a PRRSV-ADE infection may have inhibited the JAK-STAT signals to block the expression of antiviral proteins by various ways. 

Notably, multiple antiviral proteins were also dramatically decreased in PAMs during the PRRSV-ADE infection, including ISG15, ISG20, IFIT1 (ISG56), IFIT2 (ISG54), IFIT3 (ISG60), IFIT5 (ISG58), OAS1, Mx1, RSAD2, TRIM22, TRIM25 and TRIM34, except for TRIM52. Remarkably, among these ISGs, Mx1 can block the transcription of viral RNA in the nucleus to inhibit the virus’ proliferation [23]. ISG15, TRIM22, TRIM25 and TRIM34 can exert an antiviral activity by targeting and degrading the viral structural or nonstructural proteins [24,25,26,27]. Zhao et al. found that porcine OAS1 can also restrain the proliferation of PRRSV in vitro in the RNase L-independent way [28]. IFIT1, IFIT2, IFIT3, IFIT5 and ISG20 can inhibit the virus’ proliferation by directly degrading viral RNA, modifying viral transcripts or activating the innate immune responses [29,30,31,32]. RSAD2, as an antiviral protein, could restrict the amplification of DNA or RNA viruses by blocking the positive-sense RNA amplification during the early stages of the virus infection, targeting viral nonstructural proteins to promote its proteasomes degradation, or disrupting the lipid rafts [33,34,35]. Above all, the results indicated that a PRRSV-ADE infection might facilitate a viral replication in the PAMs through decreasing the expression of multiple antiviral proteins, which interdicted multiple stages of the viral life cycle during the virus infection.

### 4.3. PRRSV-ADE Infection Interferes the Ubiquitin–Proteasome System

Pang et al. recently reported that the ubiquitin–proteasome system (UPS) might be critical for the middle stages of the PRRSV lifecycle [36]. However, we found that the expression levels of 19 proteasome subunit components (PSMA1, PSMA2, PSMA5, PSMB3, PSMB5, PSMB6, PSMB7, PSMB9, PSMC1, PSMC2, PSMC6, PSMD12, PSMD13, PSMD2, PSMD4, PSMD8, PSMD9, PSME1 and PSME3) were dramatically decreased in the PAMs of the early period of a PRRSV-ADE infection (Figure 6). Interestingly, multiple studies showed that antibodies targeting the nucleoprotein of the enveloped viruses could protect the host from a virus challenge in vivo, including arenaviruses, influenza viruses and coronaviruses [37,38,39]. Caddy et al. also found that the cytosolic Fc receptor TRIM21 could target viral nucleoprotein antibodies to mediate the nucleoprotein for cytosolic degradation and induce T cell immunity by activating the UPS [40]. Notably, Cancel-Tirado et al. found that the monoclonal antibodies against the nucleoprotein of PRRSV could significantly suppressed the virus replication in PAMs [3]. Therefore, we speculate that the inhibition of the ubiquitin–proteasome system may promote a viral proliferation in the PAMs with a PRRSV-ADE infection. 

### 4.4. PRRSV-ADE Infection Up-Regulates the Expression of Mitochondrial Respiratory Chain Complexes

The electron transport chain (ETC), which is composed of five multi-subunit complexes (mitochondrial respiratory chain complexes I-V), is the main site of ATP synthesis in mitochondria. We found that the expression levels of the mitochondrial respiratory chain complexes were significantly increased in the PAMs infected with PRRSV-ADE in this study, including complex I (including NDUFA11, NDUFA12, NDUFA4, NDUFA7, NDUFAB1, NDUFAF4, NDUFAF7, NDUFB1, NDUFB4, NDUFB7, NDUFB9, NDUFC2, NDUFS1, NDUFS5, NDUFS6 and NDUFS7), III (including UQCR10, UQCRB, UQCRC1, UQCRC2 and UQCRQ), IV (including COX-5B, COX6-A1, COX-6B, COX-6C and COX-7C) and V (including ATP5F1D, ATP5MC3, ATP5ME, ATP5ME, ATP5MG, ATP6V0C, ATP6V0D1, ATP6V1D and ATP6V1H) (Figure 6). Similarly, Chan et al. also reported that the mRNA levels of the components of the mitochondrial respiratory chain complexes, especially complex I, were significantly up-regulated during the DENV-ADE infection, and which might favor the DENV replication [9]. Qu et al. also found that the complexes III sustain the replication of the hepatitis E virus (HEV) in an ATP independent [41]. Moreover, previous studies have shown that a PRRSV infection causes mitochondria dysfunction, leads to the collapse of the mitochondrial trans-membrane potential and stimulates the production of reactive oxygen species (ROS) [42]. However, COX5B, as a member of the cytochrome c oxidase complex, can decreased MAVS-mediated antiviral signaling and the ROS levels in host cells [19]. Hence, the results suggested that the expression of the mitochondrial respiratory chain complexes may enhance the replication of the virus during the PRRSV-ADE infection, and the regulatory mechanism needs to be further investigated. 

### 4.5. PRRSV-ADE Infection Interferes the Function of the Ribosome

Viruses can hijack a wide range of host ribosomal proteins to support the life cycle of themselves [43]. In this study, we found 35 differentially expressed ribosomal proteins (RPs) in the PAM with a PRRSV-ADE infection. Additionally, among them, the expression levels of all the rest of the RPs were significantly down-regulated except for RPS27A (Figure 6). Among these RPs, RPS27A can enhance the Epstein–Barr virus (EBV) proliferation and invasion by stabilizing the latent membrane protein 1 (LMP1) of the EBV and interacting directly with it [44]. RPS2, RPL6, RPL9, RPS8, RPS16, RPS17, RPL23, RPL26 and RPL27 could also increase or participate in the reproduction of the virus by different strategies, such as regulating the viral nucleic acid replication, gene transcription, viral mRNA translation initiation, etc. [43]. However, several RPs, including RPS20, RPL13A, RPL13 and RPS3, also could inhibit the proliferation of viruses by activating the innate immune signaling or suppressing the translation of a specific viral mRNA [43,45,46]. Interestingly, Guan et al. found that Foot-and-mouth disease virus (FMDV) could reduce the antiviral activity of RPL13 to a minimum by degrading RPL13, and the remaining amount of RPL13 were enough to sustain a viral replication [45]. Above all, we assume that the PRRSV-ADE infection may antagonize the antiviral activity of the host to facilitate the virus’ proliferation through reducing the expression of RPs, and the low expression levels of these RPs are still sufficient to sustain the virus’ proliferation. 

In conclusion, this study first explored the proteome alterations of PRRSV-ADE-infected PAMs using TMT-LC-MS/MS. After analyzing the DEPs of the PAMs, we found that a PRRSV-ADE infection significantly increased the expression of mitochondrial respiratory chain complexes, and interfered the functions of the innate immune signaling, the antiviral proteins, the ubiquitin–proteasome system and the ribosome. These findings would support many opportunities to elucidate the mechanisms of a PRRSV-ADE infection, screen the novel targets for the therapy of an ADE infection and develop the novel PRRSV vaccines.

## Figures and Tables

**Figure 1 viruses-15-00036-f001:**
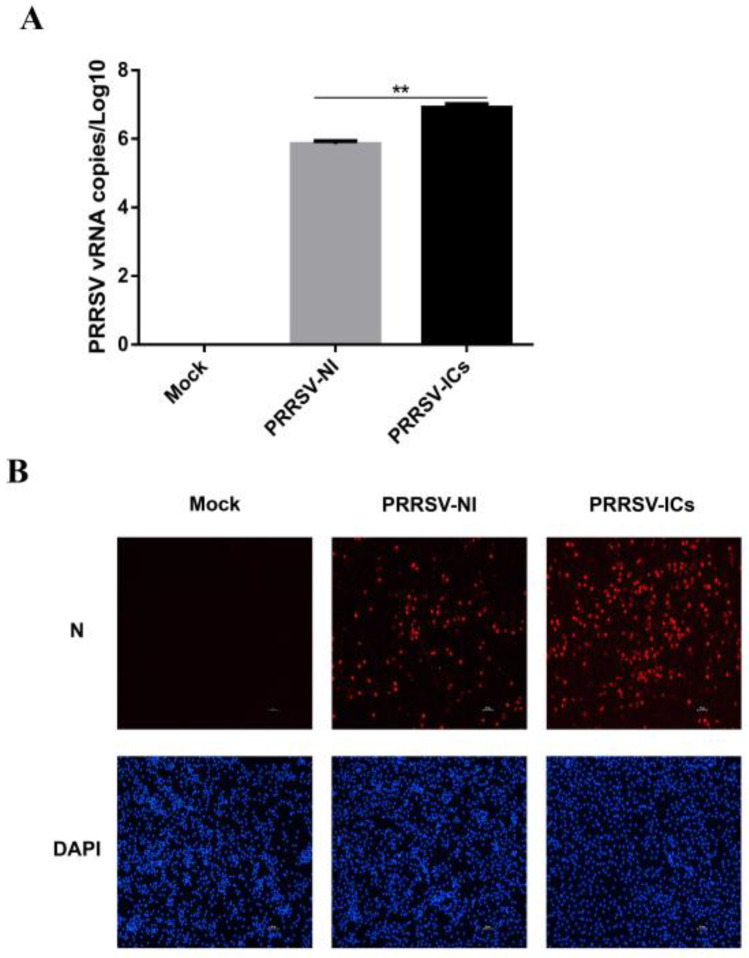
The establishment of the PAMs model with PRRSV-ADE infection. (**A**) viral loads of the cell culture supernatants in different infection groups were monitored by RT-qPCR. (**B**) Confirmation of the PRRSV proliferation in PAMs of the mock group, the PRRSV-NI group and the PRRSV-ICs group by IFA. The N protein (red) and the nucleus (blue) of these PAMs were counterstained. Scale bar = 50 μm. ** *p* < 0.01.

**Figure 2 viruses-15-00036-f002:**
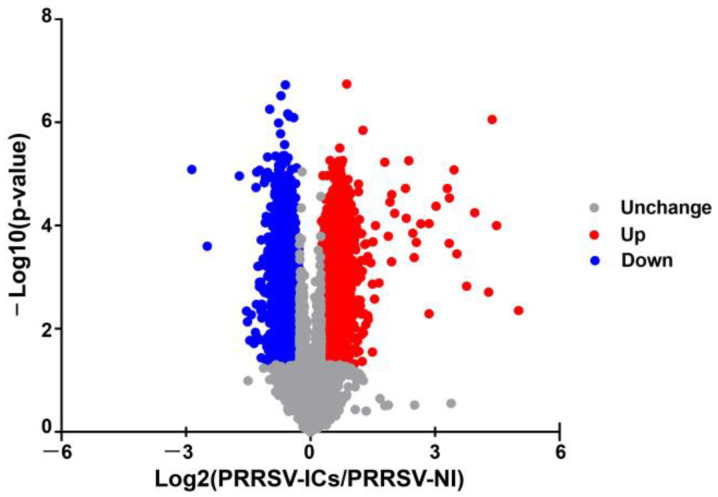
The volcano plot of DEPs in PAMs of the PRRSV-NI group and the PRRSV-ICs group. Upregulated proteins are shown in red, downregulated proteins are shown in blue and unchanged proteins are shown in gray.

**Figure 3 viruses-15-00036-f003:**
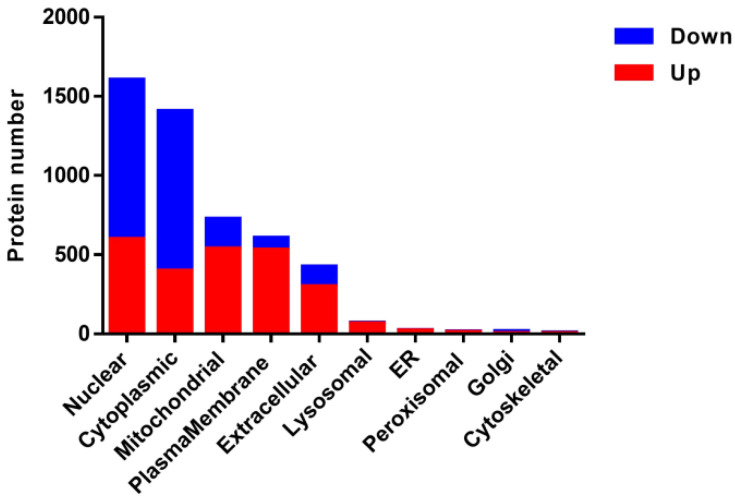
Subcellular annotation of the DEPs in PAMs infected with PRRSV-ADE infection. Upregulated proteins are shown in red, and downregulated proteins are shown in blue.

**Figure 4 viruses-15-00036-f004:**
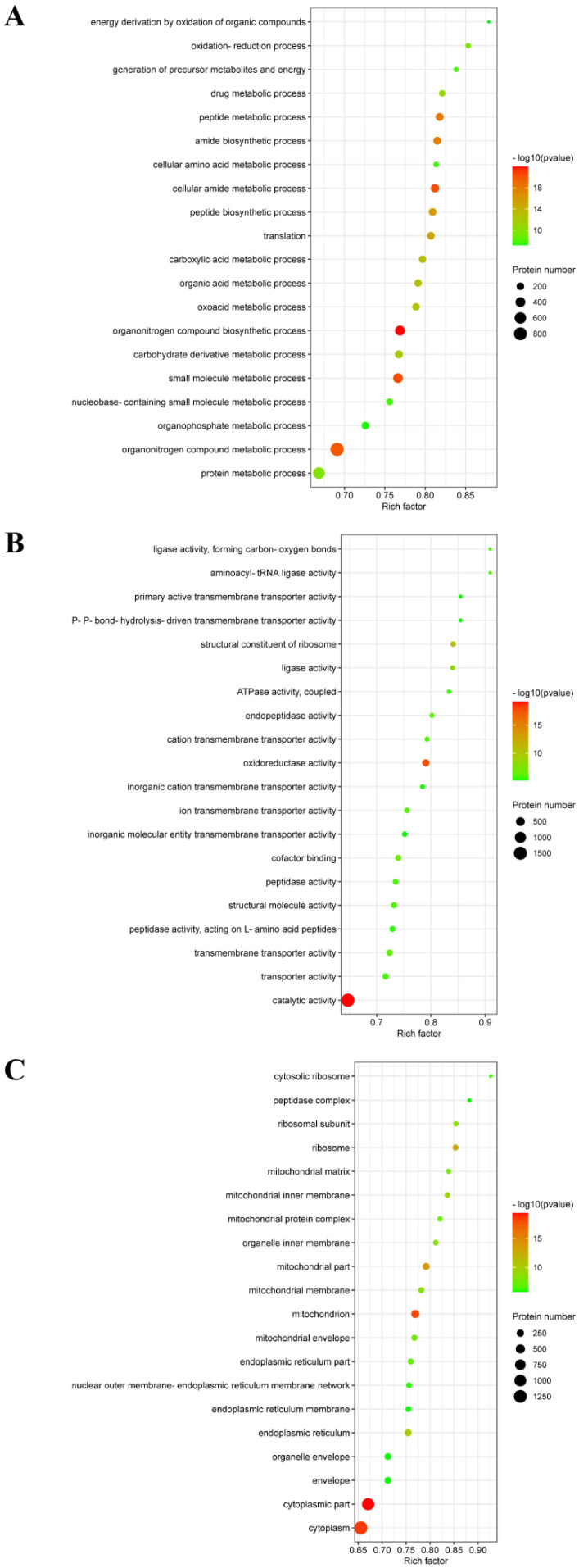
GO enrichment analysis of the DEPs in PAMs with PRRSV-ADE infection for biological process (**A**), molecular function (**B**) and cellular component (**C**). The abscissa in the figure is the rich factor which is the ratio of the number of different proteins in the corresponding pathway to the total number of proteins identified by the pathway. The color of the dot represents the *p* value of the hypergeometric test. The color ranges from green to red. The redder the color is, the smaller the value is. The size of the point represents the number of differential proteins in the corresponding pathway.

**Figure 5 viruses-15-00036-f005:**
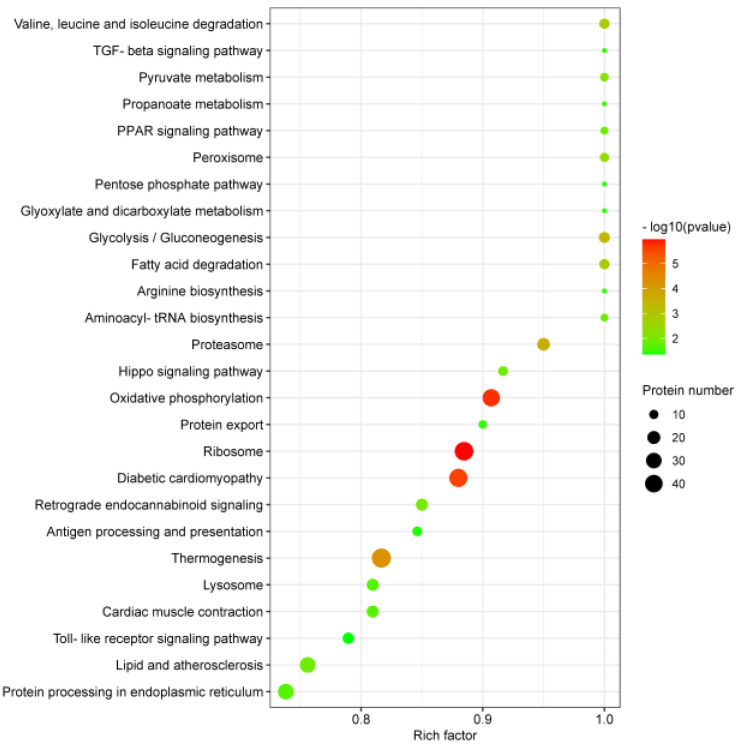
KEGG enrichment analysis of the DEPs in PAMs with PRRSV-ADE infection. Rich factor refers to the ratio of DEPs annotated in this pathway to all identified protein numbers annotated in this pathway. A larger rich factor with a lesser *p* value indicates a greater intensiveness. The top 26 enriched pathway terms were revealed.

**Figure 6 viruses-15-00036-f006:**
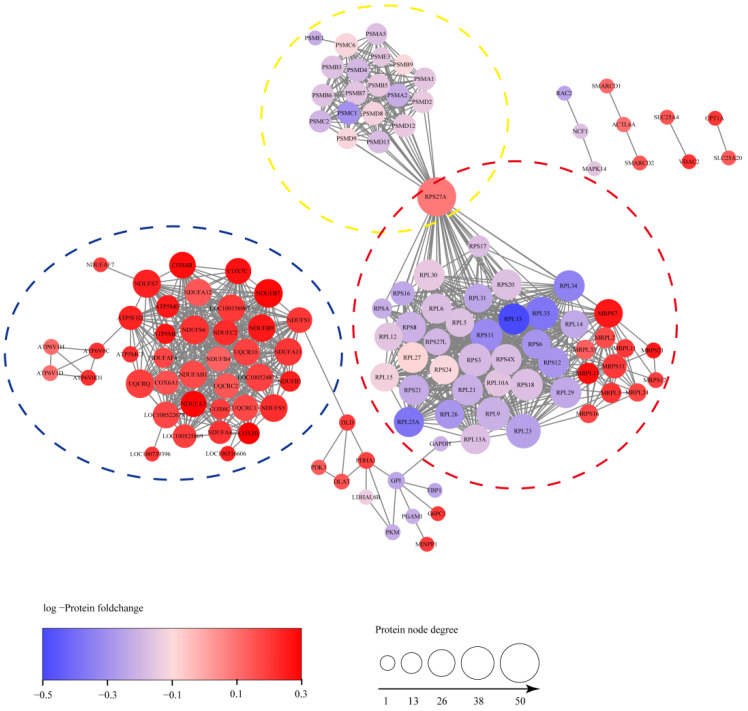
The PPI network of 143 DEPs. The connections between proteins indicate that they interact with each other and the confidence interval is greater than 0.9. DEPs were mainly concentrated in three PPI network clusters including ribosome, proteasome and mitochondria, which were circled with red, yellow and blue, respectively.

**Figure 7 viruses-15-00036-f007:**
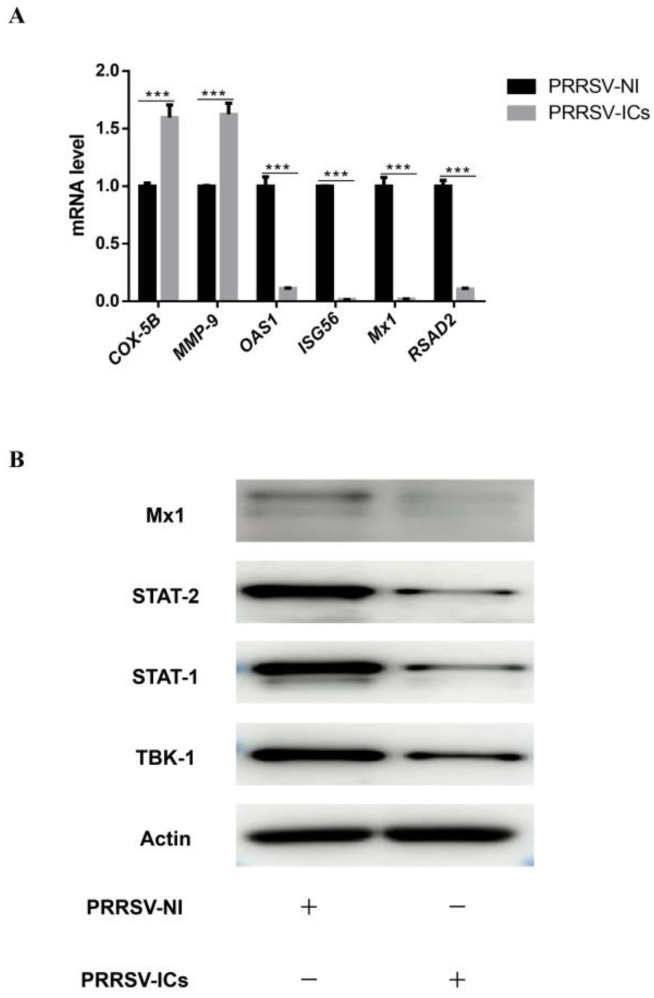
The validation of the DEPs in PAMs with PRRSV-ADE infection. (**A**) RT-qPCR analysis of six selected DEPs (*COX-5B*, *MMP-9*, *ISG56*, *OAS1*, *Mx1* and *RSAD2*) in PAMs with PRRSV-ADE infection. (**B**) Western blot analysis of four DEPs (TBK-1, STAT-1, STAT-2 and Mx1) in PAMs with PRRSV-ADE infection. *** *p* < 0.001.

**Table 1 viruses-15-00036-t001:** Forty DEPs associated with antiviral innate immune signals identified in this study.

Accession	Protein Name	Gene Name	Fold Change (PRRSV-ICs/PRRSV-NI)	*p* Value
A0A0B8RS69	Tripartite motif-containing protein 34 (TRIM34)	*TRIM34*	0.700558702	0.008236
A0A0B8RT27	Toll-like receptor 3 (TLR3)	*TLR3*	1.480082142	0.000012
A0A0B8S066	Interferon regulatory factor 9 (IRF9)	*IRF9*	0.645852502	0.001426
A0A286ZN85	Signal transducer and activator of transcription 1 (STAT-1)	*STAT-1*	0.679089935	0.000960
A0A287A5V9	DEAD-box protein 6 (DDX6)	*DDX6*	0.577778435	0.002041
A0A287AF06	Interleukin-1 receptor-associated kinase-like 2 (IRAK2)	*IRAK2*	0.707394609	0.016606
A0A287ANQ6	Tripartite motif-containing protein 52 (TRIM52)	*TRIM52*	1.579322218	0.001208
A0A287AVQ1	DEAD-box protein 3X (DDX3X)	*DDX3X*	0.634828774	0.000292
A0A385XIH5	2’-5’ oligoadenylate synthase 1 (OAS1)	*OAS1*	0.643730896	0.002701
A0A4X1SHM4	Tripartite motif-containing protein 25 (TRIM25)	*TRIM25*	0.664125491	0.002789
A0A4X1SM65	DExH-box protein 29 (DHX29)	*DHX29*	0.722151262	0.000626
A0A4X1SSZ9	DExH-box protein 36 (DHX36)	*DHX36*	0.746876328	0.010250
A0A4X1TQG4	TNF receptor-associated factor 3 (TRAF3)	*TRAF3*	0.79529251	0.010987
A0A4X1TZR6	Signal transducer and activator of transcription 3 (STAT-3)	*STAT-3*	0.588242937	0.001812
A0A4X1U0W0	Matrix metalloproteinase-9 (MMP-9)	*MMP-9*	2.034346704	0.000695
A0A4X1UBG5	DEAD-box protein 1 (DDX1)	*DDX1*	0.717280944	0.000119
A0A4X1W175	Signal transducer and activator of transcription 6 (STAT-6)	*STAT-6*	0.757959754	0.000284
A0A4X1W5D6	TANK-binding kinase 1 (TBK-1)	*TBK-1*	0.817144901	0.022810
A0A4X1W8B3	Signal transducer and activator of transcription 2 (STAT-2)	*STAT-2*	0.702062421	0.000623
A0A5G2QC44	Tripartite motif-containing protein 21 (TRIM21)	*TRIM21*	0.675237803	0.000027
A0A5G2R0A9	Interferon-induced protein with tetratricopeptide repeats 5 (IFIT5)	*IFIT5*	0.553411483	0.000016
A2TF48	Myeloid differentiation primary response gene 88 (MyD88)	*MyD88*	0.605853551	0.022306
A9QT42	Interleukin-1 receptor-associated kinase 4 (IRAK4)	*IRAK4*	0.64810251	0.000538
B2ZDZ2	Interferon-stimulated gene 15 (ISG15)	*ISG15*	0.416379988	0.000615
B3XXC2	Toll-like receptor 8 (TLR8)	*TLR8*	1.577213769	0.000043
B6ICV1	Tripartite motif-containing protein 26 (TRIM26)	*TRIM26*	0.630737025	0.007867
D4PAR3	Toll-like receptor 4 (TLR4)	*TLR4*	1.493009896	0.003026
F1RLV7	Tripartite motif-containing protein 22 (TRIM22)	*LOC733579*	0.713581442	0.000506
F1S5A8	DExH-box protein 15 (DHX15)	*DHX15*	1.234949706	0.018624
F1SCY2	Interferon-induced protein with tetratricopeptide repeats 3 (IFIT3)/Interferon-stimulated gene 60 (ISG60)	*IFIT3/ISG60*	0.426091944	0.000606
I3L5Z9	Zinc finger NFX1-type containing 1 (ZNFX1)	*ZNFX1*	0.408049746	0.000009
I3LB04	Interferon-stimulated gene 20 (ISG20)	*ISG20*	0.519256582	0.004996
J7FIC7	Interferon-induced protein with tetratricopeptide repeats 1 (IFIT1)/Interferon-stimulated gene 56 (ISG56)	*IFIT1/ISG56*	0.462410477	0.000015
J7FJH8	Interferon-induced protein with tetratricopeptide repeats 2 (IFIT2)/Interferon-stimulated gene 54 (ISG54)	*IFIT2/ISG54*	0.551217726	0.004211
K7GND3	DExH-box protein 9 (DHX9)	*DHX9*	1.509146168	0.003557
K7GS53	DExH-box protein 58 (DHX58)	*DHX58*	0.428630681	0.000008
Q59HI8	Toll-like receptor 2 (TLR2)	*TLR2*	1.24016114	0.027161
Q5S3G4	Cytochrome coxidase subunit 5B (COX-5B)	*COX-5B*	2.101483516	0.000243
Q9MZU4	Radical S-adenosyl methionine domain-containing protein 2 (RSAD2)	*RSAD2*	0.13795664	0.000008
X2KPB3	Interferon regulatory factor 7 (IRF7)	*IRF7*	0.373258619	0.005359

## Data Availability

The original data reported in this paper have been deposited in the OMIX, China National Center for Bioinformation/Beijing Institute of Genomics, Chinese Academy of Sciences (https://ngdc.cncb.ac.cn/omix: accession no. OMIX002135, accessed on 20 December 2022).

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
