# Peer review of "Proteomic Characterization of PAMs with PRRSV-ADE Infection"

_viruses, 2022, doi:10.3390/v15010036_

Round 1

Reviewer 1 Report

Comments to the Author

PRRSV-ADE infection is one of the important characteristics of PRRSV infection, and it brought great difficulties to control PRRS. In this manuscript, the authors investigated the proteomic characterization of PAMs with PRRSV-ADE infection, and identified 3935 differentially expressed proteins. Bioinformatical tools were used to analysis the proteins. Authors verified several differentially expressed proteins by RT-qPCR or Western bloting. This manuscript is clearly written with some minor mistakes. With some clarifications noted below as comments, this work will be of interest to some readers studying PRRSV-ADE infection and PRRS prevention and control.

1. Please supplement the information about anti-PRRSV IgG, such as ELISA titer, the purification method in Materials and methods section.

2. What were the working concentrations of anti-PRRSV IgG, negative IgG and the virus?

3. Why was the proteomic analysis only on the 6h infected cells?

4. Please add scale bar in Figure 1B?

5. The relationship between PRRSV infection and mitochondria is unclear. How was the mitochondrion affecting the PRRSV-ADE phenomenon? Please discuss it in discussion.

Author Response

Dear Reviewer:

Thank you for your comments on our manuscript, entitled “Proteomic Characterization of PAMs with PRRSV-ADE Infection” (Manuscript ID: viruses-2106570). They are all valuable and very helpful for revising and improving our paper, as well as the important guiding significance to our researches. According to your comments, we thoroughly revised our manuscript. The revised contents were marked up using the “Track Changes” function. The point-by-point responses to your comments are as follows:

Point 1: Please supplement the information about anti-PRRSV IgG, such as ELISA titer, the purification method in Materials and methods section.

Response 1:  Thank you for your suggestion. We have revised and supplemented the information (ELISA titer, the purification method) about anti-PRRSV IgG. Please see Line 71-72 in revised manuscript.

Point 2: What were the working concentrations of anti-PRRSV IgG, negative IgG and the virus?

Response 2: Thank you for your suggestion. We have supplemented the information (the working concentrations of anti-PRRSV IgG, negative IgG and the virus). Please see Line 76-77 in revised manuscript.

Point 3: Why was the proteomic analysis only on the 6h infected cells?

Response 3: Thanks for your questions. Previous studies have shown that Fc receptor-mediated endocytosis could promote the attachment and internalization of the virus-antibody immune complexes into the host cells, and the PRRSV proliferation [1-3]. But the differences between PRRSV and PRRSV-ADE infection is not yet clear. Thus, the manuscript aims to explore the proteomic characterization of PAMs at the early stage of PRRSV-ADE infection, which maybe provide some new ideas for the mechanism of virus-antibody-host interactions. So, we  only performed the proteomic analysis of PAMs at 6 hpi in this study.    

[1] Shi, P.; Zhang, L.; Wang, J.; Lu, D.; Li, Y.; Ren, J.; Shen, M.; Zhang, L.; Huang, J. Porcine FcepsilonRI Mediates Porcine Reproductive and Respiratory Syndrome Virus Multiplication and Regulates the Inflammatory Reaction. Virol Sin. 2018, 33, 249-260. https://doi.org/10.1007/s12250-018-0032-3.

[2] Wan, B.; Chen, X.; Li, Y.; Pang, M.; Chen, H.; Nie, X.; Pan, Y.; Qiao, S.; Bao, D. Porcine FcgammaRIIb mediated PRRSV ADE infection through inhibiting IFN-beta by cytoplasmic inhibitory signal transduction. Int J Biol Macromol. 2019, 138, 198-206. https://doi.org/10.1016/j.ijbiomac.2019.07.005.

[3] Zhang, L.; Li, W.; Sun, Y.; Kong, L.; Xu, P.; Xia, P.; Zhang, G. Antibody-Mediated Porcine Reproductive and Respiratory Syndrome Virus Infection Downregulates the Production of Interferon-alpha and Tumor Necrosis Factor-alpha in Porcine Alveolar Macrophages via Fc Gamma Receptor I and III. Viruses. 2020, 12. https://doi.org/10.3390/v12020187. 

Point 4: Please add scale bar in Figure 1B?

Response 4: Thank you for your suggestion. We have supplemented the scale bar of Figure 1B (Line 182) in revised manuscript.

Point 5: The relationship between PRRSV infection and mitochondria is unclear. How was the mitochondrion affecting the PRRSV-ADE phenomenon? Please discuss it in discussion.

Response 5: Thanks for your questions. Previous studies have shown that PRRSV infection causes mitochondria dysfunction, leads a collapse of the mitochondrial trans-membrane potential and stimulates the production of reactive oxygen species (ROS) [4]. But, COX5B, as a member of the cytochrome c oxidase complex, can decreased MAVS-mediated antiviral signaling and ROS levels in host cells [5]. We add this contents in Discussion section. Please see Line 389-391 in revised manuscript.

[4] Lee, S. M.; Kleiboeker, S. B. Porcine reproductive and respiratory syndrome virus induces apoptosis through a mitochondria-mediated pathway. Virology. 2007, 365, 419-434. https://doi.org/10.1016/j.virol.2007.04.001

[5] 19.Jacobs, J. L.; Coyne, C. B. Mechanisms of MAVS regulation at the mitochondrial membrane. J Mol Biol. 2013, 425, 5009-5019. https://doi.org/10.1016/j.jmb.2013.10.007.

Reviewer 2 Report

PRRS is still one of the major infectious diseases concerned by the pig industry. The PRRSV-ADE infection is likely to be a great obstacle for selection of immune strategies and development of high-efficiency PRRSV vaccines, but the changes of host proteins in PRRSV-ADE infected cells are unclear. In this manuscript, authors have found 3935 differentially expressed proteins in PRRSV-ADE cells, which were involved in antiviral innate immune, ubiquitin-proteasome system, mitochondrial respiratory, and ribosome innate immune. The results are valuable for scientists interested in PRRSV-ADE, and provide new research direction for ADE infection mechanism. But there are also some questions as follows:

1. Why to decide the viral infection with 6h? And what’s the titers in NI and ICs groups?

2. The authors have used GO Ontology and KEGG pathway analysis to analyze the differentially expressed proteins, but there’s no relative discussion, please add.

3. It is necessary to describe in manuscript why the strain (strain HeN-3 (GenBank ID: ON645930) was used to infect, since the HP-PRRSV infection may result in different conclusion.

4. The use of punctuation marks is not accurate. Please proofread the punctuation marks in the context. E.g. “plasma-membrane, and extracellular.” in 3.3 section, should be “plasma-membrane and extracellular”; “But the molecular mechanisms via Fc Rs in PRRSV-ADE infection, have not yet to be precisely elucidated...” in introduction, should delete the “,”.

5. The manuscript should be checked for grammar and spelling mistakes. Some sentences in the text are too long, making it difficult to understand. For example, Lines 46, " But the molecular mechanisms via Fc Rs in PRRSV-ADE infection "; Lines 230-231, " and they were was mainly involved in ..." should be " and they were mainly involved in..."; The legend of Figure 1, “(B) The efficiency of PRRSV in PAMs of different groups were detected by IFA.” ……Please correct carefully.

6. Lines 140, 307. "TBK1", "SHP-1" appeared for the first time, please write the full names of these abbreviations. Please check carefully abbreviations in manuscripts.

7. Lines 169, 274. Please replace qRT-PCR with RT-qPCR.

8.  Lines 125. "1h", "10 minutes". Please unify the description of time.

Author Response

Dear Reviewer:

Thank you for your comments on our manuscript, entitled “Proteomic Characterization of PAMs with PRRSV-ADE Infection” (Manuscript ID: viruses-2106570). They are all valuable and very helpful for revising and improving our paper, as well as the important guiding significance to our researches. According to your comments, we thoroughly revised our manuscript. The revised contents were marked up using the “Track Changes” function. The point-by-point responses to your comments are as follows:

Point 1: (1) Why to decide the viral infection with 6h? (2) And what’s the titers in NI and ICs groups? 

Response 1: Thanks for your questions. (1) Thanks for your questions. Previous studies have shown that Fc receptor-mediated endocytosis could promote the attachment and internalization of the virus-antibody immune complexes into the host cells, and the PRRSV proliferation [1-3]. But the differences between PRRSV and PRRSV-ADE infection is not yet clear. Thus, the manuscript aims to explore the proteomic characterization of PAMs at the early stage of PRRSV-ADE infection, which maybe provide some new ideas for the mechanism of virus-antibody-host interactions. So, we  only performed the proteomic analysis of PAMs at 6 hpi in this study. (2) We have supplemented the information about the titers in NI and ICs groups. Please see Line 76-77 in revised manuscript.     

[1] Shi, P.; Zhang, L.; Wang, J.; Lu, D.; Li, Y.; Ren, J.; Shen, M.; Zhang, L.; Huang, J. Porcine FcepsilonRI Mediates Porcine Reproductive and Respiratory Syndrome Virus Multiplication and Regulates the Inflammatory Reaction. Virol Sin. 2018, 33, 249-260. https://doi.org/10.1007/s12250-018-0032-3.

[2] Wan, B.; Chen, X.; Li, Y.; Pang, M.; Chen, H.; Nie, X.; Pan, Y.; Qiao, S.; Bao, D. Porcine FcgammaRIIb mediated PRRSV ADE infection through inhibiting IFN-beta by cytoplasmic inhibitory signal transduction. Int J Biol Macromol. 2019, 138, 198-206. https://doi.org/10.1016/j.ijbiomac.2019.07.005.

[3] Zhang, L.; Li, W.; Sun, Y.; Kong, L.; Xu, P.; Xia, P.; Zhang, G. Antibody-Mediated Porcine Reproductive and Respiratory Syndrome Virus Infection Downregulates the Production of Interferon-alpha and Tumor Necrosis Factor-alpha in Porcine Alveolar Macrophages via Fc Gamma Receptor I and III. Viruses. 2020, 12. https://doi.org/10.3390/v12020187. 

Point 2: The authors have used GO Ontology and KEGG pathway analysis to analyze the differentially expressed proteins, but there’s no relative discussion, please add. 

Response 2: Thank you for your suggestion. In this study, GO Ontology and KEGG pathway analysis revealed that there was a higher correlation between PRRSV-ADE infection and the top 6 KEGG pathways (ribosome, oxidative phosphorylation, diabetic cardiomyopathy, thermogenesis, proteasome and glycolysis/gluconeogenesis). Then, analysis of the Protein-Protein Interactions (PPI) network has shown that all differentially expressed proteins (DEPs) enriched in the top 6 KEGG pathways were mainly distributed in three clusters (proteasome, ribosome and mitochondria) (Line 360-362, 379-384). So it also indicated PRRSV-ADE infection might differentially regulate the functions of ribosome, proteasome and mitochondria to promote virus proliferation. Thus, we mainly analyzed the functional changes of ribosome, proteasome and mitochondria during PRRSV-ADE infection in Discussion section. Moreover, we also found that some (COX-5B, RPL13, RPS2, RPL6, RPL9, RPS8, RPS16, RPS17, RPL23, RPL26, RPL27, RPS20, RPL13A, RPL13 and RPS3) of the DEPs in three clusters could affect virus replication or natural immune response through literature review, and analyzed the relationship between PRRSV infection and RPs (Line 401-412). Finaly, we have analyzed the relationship between PRRSV infection and COX-5B (Line 389-393) in revised manuscript .

Point 3: It is necessary to describe in manuscript why the strain (strain HeN-3 (GenBank ID: ON645930) was used to infect, since the HP-PRRSV infection may result in different conclusion. 

Response 3: Thanks for your questions. Firstly, PRRSV VR-2332 like strains (the typical North American) are commonly used to study the mechanism of PRRSV-ADE infection, such as VR-2332 ISU-P and BJ-4 [4-6]. The HeN-3 strain (GenBank ID: ON645930) used in this study is also a PRRSV VR-2332 like strain. Secondly, this study aims to explore the proteomic characterization of PAMs with PRRSV-ADE infection, but not PRRSV infection. The ADE effect of PRRSV infection is that Fc receptor-mediated endocytosis promotes the attachment and internalization of the virus-antibody immune complexes into the host cells and the PRRSV proliferation [7-9]. Thirdly, no study until now had shown there were differences between the ADE effect of PRRSV VR-2332 infection and of HP-PRRSV infection. Therefore, we used the HeN-3 strain to research the PRRSV-ADE infection.

[4] Christianson, W. T.; Choi, C. S., Collins, J. E.; Molitor, T. W.; Morrison, R. B.; Joo, H. S. (1993). Pathogenesis of porcine reproductive and respiratory syndrome virus infection in mid-gestation sows and fetuses. Can J Vet Res. 1993, 57, 262-268.

[5] Qiao S.; Jiang Z.; Tian X, et al. Porcine Fcγ RIIb mediates enhancement of porcine reproductive and respiratory syndrome virus (PRRSV) infection. PLoS One. 2011. 6, e28721. https://doi.org/10.1371/journal.pone.0028721

[6] Cancel-Tirado SM, Evans RB, Yoon KJ. Monoclonal antibody analysis of porcine reproductive and respiratory syndrome virus epitopes associated with antibody-dependent enhancement and neutralization of virus infection. Vet Immunol Immunopathol. 2004. 102, 249-262. https://doi.org/10.1016/j.vetimm.

[7] Shi, P.; Zhang, L.; Wang, J.; Lu, D.; Li, Y.; Ren, J.; Shen, M.; Zhang, L.; Huang, J. Porcine FcepsilonRI Mediates Porcine Reproductive and Respiratory Syndrome Virus Multiplication and Regulates the Inflammatory Reaction. Virol Sin. 2018, 33, 249-260. https://doi.org/10.1007/s12250-018-0032-3.

[8] Wan, B.; Chen, X.; Li, Y.; Pang, M.; Chen, H.; Nie, X.; Pan, Y.; Qiao, S.; Bao, D. Porcine FcgammaRIIb mediated PRRSV ADE infection through inhibiting IFN-beta by cytoplasmic inhibitory signal transduction. Int J Biol Macromol. 2019, 138, 198-206. https://doi.org/10.1016/j.ijbiomac.2019.07.005.

[9] Zhang, L.; Li, W.; Sun, Y.; Kong, L.; Xu, P.; Xia, P.; Zhang, G. Antibody-Mediated Porcine Reproductive and Respiratory Syndrome Virus Infection Downregulates the Production of Interferon-alpha and Tumor Necrosis Factor-alpha in Porcine Alveolar Macrophages via Fc Gamma Receptor I and III. Viruses. 2020, 12. https://doi.org/10.3390/v12020187.

Point 4: The use of punctuation marks is not accurate. Please proofread the punctuation marks in the context. E.g. “plasma-membrane, and extracellular.” in 3.3 section, should be “plasma-membrane and extracellular”; “But the molecular mechanisms via Fc Rs in PRRSV-ADE infection, have not yet to be precisely elucidated...” in introduction, should delete the “,”. 

Response 4: Thank you for your suggestion. We have proofread the punctuation marks in revised manuscript. Please see Line 48-49, 205, 209, 211-212, 308 and 318-321 in revised manuscript.

Point 5: The manuscript should be checked for grammar and spelling mistakes. Some sentences in the text are too long, making it difficult to understand. For example, Lines 46, " But the molecular mechanisms via Fc Rs in PRRSV-ADE infection "; Lines 230-231, " and they were was mainly involved in ..." should be " and they were mainly involved in..."; The legend of Figure 1, “(B) The efficiency of PRRSV in PAMs of different groups were detected by IFA.” ……Please correct carefully.

Response 5: Thank you for your suggestion. We have revised grammar and spelling mistakes in revised manuscript. Please see Line 48-49, 179-182, 196, 198, 246 and 258 in revised manuscript.

Point 6:  Lines 140, 307. "TBK1", "SHP-1" appeared for the first time, please write the full names of these abbreviations. Please check carefully abbreviations in manuscripts 

Response 6: Thank you for your suggestion. We have revised the full names of these abbreviations (Line 148-152, 276-277, 285 and 323) in revised manuscript.

Point 7: Lines 169, 274. Please replace qRT-PCR with RT-qPCR. 

Response 7: Thank you for your suggestion. We have corrected two abbreviations (Line 179 and 290) in revised manuscript.

Point 8: Lines 125. "1h", "10 minutes".  Please unify the description of time. 

Response 8: Thank you for your suggestion. We have revised the description of time (Line 129 and 133) in revised manuscript.  

Reviewer 3 Report

  The manuscript by Xu et al. describes a TMT-based quantitative proteomic analysis of PAMs with PRRSV-ADE infection. This study firstly explored the proteome alterations of PRRSV-ADE infected PAMs using TMT-LC-MS/MS. After analyzing the DEPs of PAMs, they found that PRRSV-ADE infection significantly increased the expression of mitochondrial respiratory chain complexes, and interfered the functions of the innate immune signaling, the antiviral proteins, the ubiquitin-proteasome system and the ribosome. Their findings support many opportunities to elucidate the mechanisms of PRRSV-ADE infection, screen the novel targets for therapeutic of ADE infection and develop the novel PRRSV vaccines. The findings were properly presented and interpreted. The authors may consider the following minor comments before published: 

1.The image quality of figure 4 and 6 is pretty low, please provide high-definition images.

2. in vitro, or in vivo in lines 24, 54, 330, 345 and other places should be written in italics.

3. line 78, delete the space between h and pi.

4. The description of Y-axis of figure 1A can be written as: PRRSV vRNA copies/Log10.

5. Figure 2, the word “down” after the blue symbol should be written as: Down.

6. Line 230, delete the word “was”.

7. Line 241, “the for” corrects as “of”.

8. Table 1, “RNA helicase” in the Description line should have one exactly name, such as, DDX6 should be DEAD-box helicase 6.

9. Line 353-354, replace the uppercase letters with the lowercase letters.

10.Correct the format of reference 11-40.

Author Response

Dear Reviewer:

Thank you for your comments on our manuscript, entitled “Proteomic Characterization of PAMs with PRRSV-ADE Infection” (Manuscript ID: viruses-2106570). They are all valuable and very helpful for revising and improving our paper, as well as the important guiding significance to our researches. According to your comments, we thoroughly revised our manuscript. The revised contents were marked up using the “Track Changes” function. The point-by-point responses to your comments are as follows:

Point 1: The image quality of figure 4 and 6 is pretty low,  please provide high-definition images.

Response 1: Thank you for your suggestion. We have provided high-definition images in revised manuscript. Please see Line 214-217 and 249-250 in revised manuscript.

Point 2: in vitro, or in vivo in lines 24, 54, 330, 345 and other places should be written in italics.

Response 2: Thank you for your suggestion. According to your suggestion, we have corrected the font styles of “in vitro” and “in vivo” in revised manuscript. Please see Line 25, 42, 56 and 348 in revised manuscript.

Point 3: line 78, delete the space between h and pi.

Response 3: Thanks for your comment. According to your suggestion, we have deleted the space between “h” and “pi” in revised manuscript. Please see Line 83, 168 and 172 in revised manuscript.

Point 4: The description of Y-axis of figure 1A can be written as: PRRSV vRNA copies/Log10.

Response 4: Thanks for your comment. According to your suggestion, we have corrected the information in Figure 1 (Line 175-176) in revised manuscript.

Point 5: Figure 2, the word “down” after the blue symbol should be written as: Down.

Response 5: Thanks for your comment. According to your suggestion, we have corrected the information in Figure 2 (Line 189-190) in revised manuscript.  

Point 6: Line 230, delete the word “was”.

Response 6: Thanks for your comment. According to your suggestion, we have corrected the grammar mistake (Line 246) in revised manuscript.

Point 7: Line 241, “the for” corrects as “of”.

Response 7: Thanks for your comment. we have corrected the grammar mistake (Line 258) in revised manuscript.

Point 8: Table 1, “RNA helicase” in the Description line should have one exactly name, such as, DDX6 should be DEAD-box helicase 6.

Response 8: Thank you for your suggestion. According to your suggestion, We have revised and supplemented the full names of these abbreviations in the Description line. Please see Table 1 (Line 276) in Results section.

Point 9: Line 353-354, replace the uppercase letters with the lowercase letters.

Response 9: Thank you for your suggestion. According to your suggestion, we have corrected clerical errors (Line373-374) in Discussion section in revised manuscript.  

Point 10: Correct the format of reference 11-40.

Response 10: Thank you for your suggestion. According to your suggestion, we have corrected the format of reference 1-46 (Line 445-573) in revised manuscript.